# From Sport to Work? Exploring Potentials in a Moroccan Sport-for-Employability Programme

Louis Moustakas [1,*], Viviane Raub [2], Yassine Moufagued [3] and Karen Petry [1]

1 Institute for European Sport Development and Leisure Studies, German Sport University, 50933 Cologne, Germany
2 Deutsche Gesellschaft für Internationale Zusammenarbeit (GIZ), 53113 Bonn, Germany
3 TIBU Africa, Casablanca 20250, Morocco
* Correspondence: l.moustakas@dshs-koeln.de

**Abstract:** Sport for development (SFD) has become an increasingly recognised and used approach to support positive social development across several contexts and thematic areas, including as it relates to improving the employability of young people not in employment or education (NEETs). Despite this, there has been limited research in this area, and we only have a partial picture of the experiences, mechanisms, and design of sport-for-employability programmes. Responding to this, the following paper presents the results of a qualitative study on the experiences and outcomes associated with an employability-focused SFD programme based in Casablanca, Morocco. Results illustrate how the programme provided crucial support to encourage consistent engagement and that this programme offered valuable opportunities for practical experiences and recognised qualifications. Nonetheless, many participants remain in short-term or precarious employment situations. Thus, closer alignment with job market needs and engagement with employment policy issues are likely needed to support longer-term, more secure employment.

**Keywords:** sport for development; livelihoods; employment; employability; NEETs





## 1. Introduction

Sport for development (SFD) has become an increasingly recognised and used approach to support positive social development across a number of contexts and thematic areas. Broadly speaking, SFD can be defined as the intentional use of sport, play, or physical activity to support the achievement of development objectives. Perhaps most logically, SFD has been extensively connected with physical or mental health development [1] but has also emerged as a prominent tool to tackle issues such as peacebuilding [2] or education [3]. Of particular interest here, the connections between SFD and employability have grown significantly over the last decade. In particular, sport is seen as a potential hook to attract youth, not in employment, education, or training (NEETs), develop their social networks, and direct them towards opportunities for skill development [4,5]. Today, employability represents one of the main thematic areas of SFD organisations [6], and employability has been seized upon as a core area for many international governmental agencies employing sport to support social development outcomes [7]. Likewise, an increasing array of national policies explicitly recognise the linkages between sport, economic development, and employment [8,9].

Contrasting this practical and political activity, to date, there has been limited research regarding the contribution of SFD programmes to (youth) employability [10], and there are limited insights into the workings of existing programmes [11]. Though there have been emerging projects seeking to formalise the connections between SFD and employability [5,12], we still do not have a clear picture of the activities, mechanisms, or outcomes of sport-for-employability programmes [11]. Although the currently available literature provides valuable insights into programmes, there remains a need for "in-depth insight into

the workings" of particular programmes [11]. As such, the present paper uses a qualitative approach to investigate the experiences and outcomes associated with an employability-focused SFD programme based in Casablanca, Morocco. This study took place against the background of an impact evaluation conducted on behalf of the German Development Corporation (GIZ), which actively supported the sport-for-employability initiative. In particular, three main questions drove this research: (1) what opportunities did the programme generate for participants; (2) what challenges did programme participants face, both within the programme and in the employment market; and (3) what recommendations can be made for future sport-for-employability programmes.

Moving forward, the following paper advances in five steps. First, a short overview of the definition of employability and its potential connections to sport will be provided. Second, we will present the general background of the programme in question. Third, details of the methodology used in this study are described, and fourth, the key themes developed will be presented in the results. Finally, the results will be contextualised to provide further recommendations for research and practice.

## 2. Employability and the Role of Sport

The term employability has evaded consistent definition at the academic and policy levels [4,13] but generally refers to how an individual's skills, knowledge, or characteristics enable them to find and maintain employment [4,5,13]. For instance, in the Higher Education context, it has been defined as "a set of achievements, understandings and personal attributes that make individuals more likely to gain employment and to be successful in their chosen occupations" [14]. Flowing from this, policy actions and initiatives tend to concentrate on developing or strengthening components related to personal skills and attributes. For instance, the International Labour Organisation (ILO) highlights the importance of education, training, and soft skills such as teamwork, problem solving, and communication as essential for employability [15]. Likewise, as the labour market has generally shifted to more non-manual work and put a premium on soft skills, the focus on these individual traits has further grown in relation to employability [16].

Many authors, however, have criticised this individual, or supply-side, focused approach, as it shifts blame to jobless (young) individuals for their predicament, as opposed to acknowledging a lack of opportunity and support in the labour market itself [17]. Indeed, recent work has shown how unfavourable conditions, such as those imposed by inequality, poverty, or inadequate policies, can drive unemployment [18,19]. Recognising these criticisms, McQuaid and Lindsay [13] argue that employability, or the ability to find or maintain employment, is contingent on more than just individual attributes. As they argue, many prominent definitions and models tend to focus uniquely on "supply-side" characteristics and ignore or minimise demand or contextual factors. As such, for them, the trait of being employable is determined by a number of personal and external factors that mutually interact with each other. As illustrated and summarised in Table 1, these can be broadly categorised into individual factors, personal factors, and external factors.

**Table 1.** Broad model of employability, adapted from [13].

| Individual Factors | Personal Factors | External Factors |
|---|---|---|
| <ul><li>Employability skills and attributes (e.g., personal competences, transferable skills, qualifications)</li><li>Demographic characteristics</li><li>Health and well-being</li><li>Job-seeking skills</li><li>Adaptability and mobility</li></ul> | <ul><li>Household circumstances (e.g., care responsibilities, housing quality)</li><li>Work culture (e.g., support of work within peer environment)</li><li>Access to resources (e.g., financial capital, transport, social capital)</li></ul> | <ul><li>Demand factors (e.g., labour market demand, job offer characteristics, etc.)</li><li>Enabling support factors (e.g., employment policy)</li></ul> |

Alongside these more general debates and programmes around employability, there has been growing recognition from governments and practitioners of the potential for sport to contribute to economic development and (youth) employability [4,8,20], including in Morocco [9,21,22]. This recognition comes from two fronts. First, there is an understanding that sport can provide an attractive and interactive setting that allows for experiential learning and the development of knowledge and skills that are seen as essential in the job market. Indeed, due to its widespread appeal as a "shared cultural manifestation", relatively low cost, and interactive nature, sport has been presented as a vehicle to support development across a wide range of areas [23,24]. In particular, sport programmes have been presented as potentially effective vehicles to develop soft skills often seen as essential in the job market [20,25]. For instance, there are long-standing claims, and some evidence, that sport can support the development of competences such as teamwork, communication, discipline, or self-confidence [25,26]. Second, the sport industry presents significant potential for economic growth. It is one of the fastest-growing industries globally, encompasses a broad range of sub-sectors, and connects to several other industries [27]. Further, policies in many countries, such as Indonesia, Botswana, and Morocco, have identified sport as a prime area for economic diversification and growth [28–30]. In short, on the one hand, sport is viewed as an effective vehicle to develop employability skills. On the other hand, sport itself is viewed as a growing industry that can be a vector for employment and economic growth.

As a result of these perceived opportunities, many SFD programmes targeting employability, both within sport and in general, have emerged, and employability is now viewed as one of the central areas for SFD practice. For instance, recent research suggests that about 17% of programmes focus on the more broadly defined area of livelihoods [6]. Typically, programmes in this area use sport as a hook to attract vulnerable youth or those not in employment or education (NEETS). Once in the programme, these organisations combine sport, skill building, and workshops to (re)direct youth towards employment or further education opportunities [4,5]. As such, these programmes can be broadly said to address individual factors related to employability, such as competences, well-being, and job-seeking skills. To some extent, due to the interactive and social nature of many of these programmes, they may also contribute to developing social relationships as well.

Despite the growing relevance of this approach, there has been limited research regarding the contribution of SFD programmes to youth employability [10], and existing research paints a mixed picture. For instance, Spaaij and colleagues [31], in a study of two European-based programmes, highlight how programmes struggle to provide stable, well-paid employment and address more structural issues around employability while still recognising the positive impacts on individual participants. Likewise, more recent work shows that some programmes may not have well-defined outcomes nor a clear concept of how a programme may contribute to those outcomes [11]. Nonetheless, the existing literature points to participants developing a variety of soft or professional skills [32–34].

On the whole, the above analysis suggests two things. First, the linkages between sport and employability have become increasingly recognised in practice and policy. Second, how sport for employability plays out in practice remains largely understudied.

## 3. Context and Structure of the SAME Programme

In Morocco, there is a need to resolve many of the challenges around economic growth and employment. Crucially, youth unemployment rates have remained a persistent challenge in the country, with numbers ranging between around 20 and 28% [21,35,36]. In turn, this unemployment is thought to contribute to several adverse social outcomes, including drug use, criminality, and decreased mental health [21,37]. To address this, both the development of infrastructures and skills are seen as necessary [21,35]. Sport, in particular, has been identified as a potential avenue to support economic development, promote innovation, and reduce unemployment [30,38].

These trends converge especially visibly in Casablanca, where the programme is based. Once a sparsely populated coastal settlement, today Casablanca is the largest city in the Maghreb, with nearly 4 million people, as well as its economic centre. This growth has been pushed by both rural migration as well as significant numbers of international migrants [39], many of whom are dispersed, hidden, and not accounted for in official statistics [40]. Much of this is driven by individuals moving to a prosperous economic centre in the pursuit of opportunities and a better life. Nonetheless, the broader trends of unemployment play out in Casablanca as well. Sport plays an important role in shaping life in the city, as Casablanca is the hub of the Moroccan sport industry, has a rich sporting history [9,41], and has increasingly seen sport positioned as a tool for further development [9,42]

Against this background, TIBU Africa works to tap into the potential of sport to support social and economic development. Based in Casablanca, TIBU Africa aims to support the education, employability, and social inclusion of youth through sport. For them, sport is seen as "an engine of personal development, allowing the acquisition of key behavioural skills (known as "life skills") and can offer employment opportunities, particularly for young people" [21]. In line with this, TIBU Africa has initiated a number of programmes around employability, life skill development, and inclusive sport participation.

The SAME programme aimed to address this challenge by supporting the integration, professional development, and employability of young migrants or returnees. This programme was designed for the benefit of Ivorian, Malian, and Senegalese nationals, as well as Moroccans returning from abroad. The objective was to allow these young people to acquire skills to be employed and foster opportunities for youth in Morocco [21,22,43].

Supported by the Ministry of Foreign Affairs, African Cooperation and Moroccans Living Abroad, SAME was co-financed by the European Union (EU) and the German Federal Ministry for Economic Cooperation and Development (BMZ). On the ground, it was implemented together by the German Development Corporation (GIZ) and TIBU. As part of this partnership with the GIZ, the German Sport University was engaged to support and implement the programme evaluation.

Through a one-year, ten-module programme, the initiative attempted to fill the gaps between skills and the demands of the labour market, as well as explore the broader potential of the sports industry. Overall, SAME supported 27 participants (25 men, two women) in 2020–2021. These participants followed a curriculum combining sport, professional, and soft skills. Modules were led by TIBU staff and external experts, and touched on various subjects, including sports, languages, management of sporting events, coaching, and SFD concepts. Of note, sport and communication modules offer the possibility of obtaining recognised fitness training (EREPS) or communication (PCM) certifications. In addition, integration into the labour market was supported via internships, mentorship, and support with self-employment activities [43]. Table 2 provides a summary of the modules.

**Table 2.** Overview of SAME modules. Adapted from [44].

| Module Number | Topic |
|---|---|
| 1 | Sport coaching (including EREPS certificate) |
| 2 | Sport for development |
| 3 | Language (e.g., English, French, Arabic) |
| 4 | Microsoft and other digital tools |
| 5 | Leadership, team coaching, and communication (including PCM certificate) |
| 6 | Sport event management |
| 7 | Sport policy in Morocco |
| 8 | Financial education |
| 9 | Sport association management |
| 10 | Social and sport entrepreneurship |

Complementing this, participants were connected to several sport industry stakeholders and supported in finding employment opportunities or developing entrepreneurial

ideas. For instance, prospective entrepreneurs could apply for TIBU's Sports Corners programme, an incubator programme where individuals participate in a six-month programme and receive support from mentors [45]. In addition, all participants were supported through a monthly stipend and other logistical assistance for the duration of the programme.

## 4. Methodology

### 4.1. Design

This study is based on a qualitative approach centred on interviews and focus group discussions with practitioners and participants of the Casablanca-based sport-for-employability programme. Additionally, observations and supporting documents were used to supplement the analysis. The qualitative design of this study was chosen to provide a more in-depth exploration of the perspectives and experiences within the programme, with a specific focus on the impact of the programme on participants. In the following sections, the exact nature of the data collection and analysis are presented.

### 4.2. Data Collection

Participants were purposively recruited from participants and stakeholders involved in TIBU's employability programmes. Connections to key informants at TIBU, as well as to GIZ staff, facilitated the majority of these interviews. In general, participants were either contacted through TIBU colleagues or directly via e-mail. In particular, data collection was greatly facilitated by the third author, who is also a senior manager at TIBU.

Before each interview, the general purpose of the research was explained to the participants, and they were assured that their statements would remain anonymous. Written consent was obtained for all interviews, and ethical approval was received from the German Sport University for the research (Application Number: 027/2022). Participants were made aware of the data protection guidelines following GIZ and German Sport University regulations, both of which respect the European General Data Protection Regulation. All but one of the interviews were recorded with a digital recorder, and participant approval was obtained to do so.

As presented in Table 3, 16 individuals (14 male, 2 female) took part in interviews or discussions. All interviews took place during a site visit by the first author in March 2022 and were staged in several different settings, including at TIBU's main educational facility, TIBU offices, and a local Higher Education institution. These interviews sought to understand their motivations for joining the programme, their experience with the SAME programme, and the perceived challenges and opportunities associated with the programme. Interviews took place in either French or English, and notes were taken following each interview. Furthermore, verbatim transcripts were produced for all interviews, except one where the participant did not consent to be recorded.

Complementing these interviews, the first author visited, observed, and interacted with TIBU activities during the visit, including football workshops, entrepreneurship events, sport activities, and site visits. In addition, as part of the M&E process, we were in regular contact with colleagues from TIBU and GIZ. This regular contact and observation gave us a first-hand glimpse of TIBU's goals, practices, and overall approach. During and following these interactions or observations, we took notes to document the physical environment, participants, exchanges, activities, and impressions and reflections. Finally, programme documents and previous internal research were used to support and contextualise the analysis.

**Table 3.** Overview of interviewees.

| Descriptor | Gender | Background | Interview Language |
|---|---|---|---|
| Participant 1 | Male | Non-Moroccan | French |
| Participant 2 | Male | Non-Moroccan | French |
| Participant 3 | Male | Non-Moroccan | English |
| Participant 4 | Male | Moroccan | French |
| Participant 5 | Male | Non-Moroccan | French |
| Participant 6 | Male | Non-Moroccan | French |
| Participant 7 | Male | Non-Moroccan | French |
| Participant 8 | Male | Non-Moroccan | French |
| Participant 9 | Male | Non-Moroccan | French |
| Staff 1 | Female | Moroccan | French |
| Staff 2 | Male | Moroccan | French |
| Staff 3 | Male | Moroccan | French |
| Staff 4 | Female | Moroccan | English |
| Staff 5 | Male | Moroccan | French |
| Staff 6 | Male | Moroccan | French |

*4.3. Data Analysis*

Analysis of interview data was performed through a process of thematic analysis (TA) [46,47]. TA offered the possibility to develop an analysis centred on the perspectives and experiences of on-the-ground stakeholders while also allowing the analysis to be informed by existing literature and theory around employability and sport.

MaxQDA 2022 was used to organise data, write memos, develop codes, and identify themes. In addition, we diarised our overall thought processes and reflections in a separate note document and tracked all interactions with participants or stakeholders in a dedicated sheet [48].

Overall, the analysis followed the six steps outlined by Braun and Clarke [46,47]. First, we familiarised ourselves with the dataset, reading and re-reading transcripts, interview notes, observation notes, and programme documents. Throughout, memos associated with specific data items and compiled thoughts, impressions, and reflections related to the entire dataset were included.

As a second step, the interview transcripts and observation notes were coded. Codes were developed inductively, though they were influenced by existing literature on sport for development, employability, and the Moroccan employment market. The codes captured a wide range of semantic (e.g., internship requirements, visa requirements) and latent (e.g., perceptions of non-Moroccans about professional opportunities) concepts.

To support the third step of theme identification, codes were merged and organised into broader categories related to outcomes, facilitators, and challenges within the sport-for-employability programme. Once this was done, review and visual mapping were undertaken to identify patterns and develop themes. Here, themes should be understood as patterns "of shared meaning organised around a central concept" [46] (p. 77). To develop these, we reviewed code excerpts and used MaxQDA's visual tools (e.g., code maps, code relations) to identify connections across the data. For steps four and five, we alternated between (re)drawing thematic maps and writing theme definitions, eventually settling on the three themes presented later in this paper. Namely, as this paper focuses specifically on outcomes and experiences in the SAME programme, we concentrated on developing themes that reflect the outcomes and experiences of programme participants. The final step involves writing the analysis, which forms the basis of the following results section.

**5. Results**

Based on the methodology above, we generated three themes to illustrate core facets of experiences within the programmes. Globally, these themes show how the programme addressed both individual and personal factors to employability while nonetheless leaving some participants in situations of continued unemployment or precarity.

In the below section, the main findings, supporting quotes, and some related analysis for each theme are presented together. As such, this should be understood as a qualitative report that mixes results and preliminary discussion and allows the results to be located in the context of research and theory [46].

*5.1. Structural Challenges and Support*

This first theme illustrates the range of structural challenges faced by participants and the efforts made by the SAME programme and staff members to help mitigate these challenges. The fact that participants face varying structural challenges should not be considered particularly surprising. After all, the target population of the SAME programme are immigrant youth not currently in education or employment. Indeed, many of the participants interviewed moved to Morocco, and Casablanca specifically, to pursue a sporting career, only to face barriers and injuries preventing them from fully integrating into professional sport: "I was a prospect there, but I got injured when I arrived. Afterwards, I went down to [lower league club] again. I didn't have a great career because of injuries" (Participant 2). Conversely, this sports background also provided an essential motivator to join the sport-focused SAME programme: "what pushed me to participate in the SAME programme is that first of all, I am interested in everything that has to do with sports" (Participant 9).

On the whole, these backgrounds meant that many participants faced precarious work situations and could not necessarily cope easily with the daily costs of living. Indeed, vulnerable groups, such as migrants, may face additional difficulties accessing labour opportunities in Morocco [36]. As such, many interviewees related stories of taking short-term or seasonal work in sectors such as shipping, agriculture, or call centres. The combination of high living costs and irregular employment meant many participants struggled to find stability or plan ahead: "we work without ever being able to save up" (Participant 1).

Given this, the stipends and personal support offered by TIBU were crucial facilitators of sustained, constructive engagement with the programme. All participants were offered a monthly stipend of MAD 2500 (approximately EUR 250)—which is just under the national minimum wage for public service employees—for the duration of the programme and assisted those who needed to find suitable accommodation: "so they took charge of accommodation and the scholarship. ( . . . ) That really helped me to follow out the programme without facing a lot of difficulties (Participant 3).

As most participants came from foreign countries, they needed support obtaining or renewing their Moroccan residence permits. In addition, due to changes brought about via the COVID-19 pandemic, obtaining this permit became more challenging, with officials suddenly requiring "new documents, new proofs" (Staff Member 6). Yet, this permit is crucial for the participants' ability to stay in the country and be considered for employment. Indeed, when applying for jobs, the first question received by many tends to be, "do you have your residence permit" (Participant 6).

To help minimise these issues, staff at TIBU helped participants secure necessary documents and directly enter into contact with local immigration offices. This support was a crucial facilitator in helping participants access the qualifications and opportunities presented by the programme. However, this support also seemed to have a certain ad hoc quality and did not appear to be initially built into the overall programme design. Namely, participants frequently highlighted how one particular staff member regularly provided support on an as-needed basis. In turn, some participants wished for more formalised and systemic support from the programme: "They need to be able to accompany the students, first with the residence permit, and after as required ( . . . ) TIBU must work in this direction" (Participant 7).

*5.2. Qualifications and Opportunities*

The regular participation in the programme facilitated by the financial and logistical support allowed participants to access several learning, certification, and practical op-

portunities, and this second theme illustrates the perceived value and experiences with these opportunities.

From an educational perspective, two core components especially stood out. First, the opportunity to access a high-quality, internationally recognised certification was a crucial added value. One of those was from the European Register of Exercise Professionals (EREPS), which is recognised worldwide as a mark of quality in the fitness area and offers many general and specialised fitness training certifications. For many, accessing this was "key", and "they were there for that" (Participant 1). Likewise, participants expressed high appreciation for the communication and leadership module, which itself was based on the Process Communication Model (PCM). As described by one staff member:

"PCM is a communication tool that allows us to discover and understand our own personality, but also the personalities of others, and allows us to develop communication strategies adapted to each type of personality. ( ... ) What is good about PCM is that it is a tool which is pragmatic and which is immediately operational. It gives you an idea of the six personality types that exist" (Staff 5).

Second, beyond obtaining qualifications, many participants highlighted how the SAME programme allowed them to improve in practice and how their new knowledge translated into their professional life. Indeed, the more practical modules and opportunities for work experience provided concrete avenues to put new knowledge into practice. For instance, some participants noted how the programme helped them improve certain aspects of their coaching: "[I could] perfect some things with the technical and training planning module" (Participant 4). Likewise, some highlighted moments where their newfound communication skills helped them manage different situations within their professional environment. One story illustrates how a participant was able to retain a player he was coaching despite that player's initial frustrations:

"There was a player who was angry, 'I won't play', but I talked with him. Now he is among the best players, but at one point, he wanted to give up. Thanks to a few things that we learned, that's what happened" (Participant 5).

Relatedly, the programme also consciously connected these certifications and the overall programme to potential opportunities for participants. For instance, TIBU maintained connections with local football academies, which provided a platform for internships or employment for the participants: "I did an internship there thanks to SAME. Then there are a lot of opportunities" (Participant 2). Within TIBU, there are also some internal opportunities to support the development of entrepreneurial ventures. For instance, participants can rent TIBU's fitness facilities at a low cost to conduct their own personal training sessions or pursue further entrepreneurial development through the Sports Orange Corners incubation programme. In short, this approach is broadly aligned with existing sport-based life skill or employability literature, which highlights the importance of practical opportunities to ensure the sustainable development of skills away from the programme setting [4,49].

Finally, it is also worth noting that for some participants, the regular and sustained relationships developed through the programme were a positive outcome of their participation. Developing connections and support allowed them to feel more comfortable and integrated within the context of their new country: "first we became a family, and then we started to integrate better" (Participant 9).

### 5.3. Moving beyond the Pitch

Despite the significant structural support and educational offers, the fact is that many of the programme participants remain in precarious employment situations. As noted in a survey of programme participants done in the context of the evaluation, as of March 2022, 46% of respondents indicated not currently being in employment, though numerous interviewees did highlight entrepreneurial or self-employment activities. However, even for those in employment, many work on part-time or irregular contracts or seek to develop as freelancers or entrepreneurs: "I just waited to 4–5 months after my internship, I've been employed, but not really officially" (Participant 3). This latter comment also reflects

challenges in the broader employment environment, whereby employment is not only scarce but often done on a short-term or informal basis, without the security of signed, long-term contracts.

To some extent, this lack of stability is also inherent to the nature of the fields (e.g., coaching, training) in which SAME participants are primarily trained to work. The need for coaches depends on demand and seasonal factors, making much of the employment irregular, unpredictable, and low-paid: "we can find ourselves lost in the job market with very low salaries that do not even exceed the SMIC [poverty line]" (Staff 1). More broadly, this reflects experiences from European programmes, whereby participants were often trained and re-oriented to further short-term or precarious employment [31,33]. This issue was also recognised internally, with one major criticism of this approach being that participants were simply taken from one form of precarious employment and trained to work in another form of precarious employment instead of encouraging participants to reach for more technical or managerial posts: "we do this programme just to retrain frontline workers to be frontline workers" (Staff 3).

Though not as trenchant or direct, this analysis can be connected to the many comments that suggest that participants need or want to be trained on a broader range of professional and creative skills, including entrepreneurial thinking, sports nutrition, sports journalism, event management, digital media, and more. Likewise, ideas of innovation and entrepreneurial competences were recurrent: "if you have the spirit, if you have the ability to innovate something that is new and different, you can do things down there" (Staff 2). By and large, these notions echo recommendations elsewhere that recognise a need to develop soft skills such as creativity, as well as hard or professional skills in related fields such as sport management [38]. For employability initiatives, in particular, these comments suggest a need to develop skills and qualifications that allow participants to aspire to employment or opportunities beyond the delivery of sport activities on the pitch or the gym.

## 6. Discussion and Conclusions

Through the three themes developed here, we have attempted to illustrate how the SAME programme provided crucial support to encourage consistent engagement with their employability programme and that this programme offered valuable opportunities for practical experiences and recognised qualifications. Nonetheless, many participants continue to find themselves in short-term or precarious employment situations, as the opportunities in question tend to reside within the low-paid, short-term field of sport or fitness coaching. In turn, further analysis of these results presents many interesting theoretical and practical implications.

Theoretically, the general approach adopted by the programme largely echoes the theory of change presented by Coalter et al. [4]. Namely, sport is used as a hook to attract youth, and these youth are brought into a supportive social environment and directed to opportunities for skill development, especially through recognised qualifications and practical experiences. Instructively, the SAME programme also illustrates the crucial importance of providing sufficient financial and administrative support, as opposed to the narrow focus on individual factors often inherent to employability programmes or policies [13]. Indeed, the support provided by SAME allowed participants to fully engage in the intensive one-year programme, which gave them a chance to access qualification and practical opportunities while minimising stressors around accommodation or cost of living. Quite simply, it is not realistic to work with NEET groups and expect the full engagement of participants while these same participants are in situations of high precarity or vulnerability. As such, future theoretical or conceptual models of sport for employability would do well to more acutely account for the importance of this kind of personal support, especially when working with vulnerable groups.

From a more practical lens, programmes themselves would benefit significantly from taking a more holistic view of employability and addressing so-called personal or external

factors. Limited examples from other programmes similarly suggest that providing such avenues for financial and administrative support can be of great value to participants [32,50]. Indeed, as highlighted above, the support provided by the SAME programme was of critical importance for participants. Despite this, SAME did not necessarily lead participants to well-paid, stable employment. Though most participants were engaged in employment, self-employment, or entrepreneurial activities, there was a certain level of precarity associated with their activities. This reflects conclusions elsewhere, which note that "approaches still remain problematic, especially with regard to the quality and longevity of employment" [31]. However, responsibility for this cannot be wholly placed with the programme, as young people being pushed towards short-term or unregulated employment is a persistent problem throughout the region [36]. In addition, there are certainly limits to what an individual programme can achieve, especially in contexts with higher unemployment [51].

We would also be remiss if we did not note the significant gender imbalance in programme participation. On the one hand, this is understandable, as youth unemployment is prevalent among males in the country [35]. On the other hand, women's labour force participation is significantly lower [52] and makes up a slight majority of migrants to the country [53], therefore making them an important target group for employability initiatives. Focusing on less male-dominated sports beyond basketball and football could provide a more suitable hook for female participants, while several additional initiatives could be considered to support their participation. For instance, a review conducted by the World Bank highlighted how childcare support and microcredit lending were particularly valuable for women in employability programmes [54].

More generally, numerous steps can be taken by programmes to minimise or improve the external factors limiting employment opportunities. Though it is unrealistic for local programmes to address demand factors such as labour market conditions or job offer characteristics, programmes can better match their activities with current realities and emerging trends. This means that programmes should align closely with the current and expected skills needed in the job market and, relatedly, support the development of high-level skills seen as important for the sector [55]. For Morocco, in particular, recent work highlights weaknesses in the development of administrative or managerial staff and a need to develop soft skills such as problem solving or creative thinking [9,36,38]. Likewise, programmes that work specifically within the sport sector should consider how to connect their participants to opportunities across the industry, including in sub-fields such as events, health, tourism, digital media, and more. As Ratten notes [27], by its nature, sport is uniquely connected to many different stakeholders and industries. Furthermore, in the Moroccan case, we echo previous recommendations to further elaborate these connections through a thorough mapping of the overall national sport sector [38].

Finally, though there is a sense that there are limits to what sport for development programmes can do in terms of social development, and employability specifically [31], there is also a need to avoid seeing the scope of such programmes as merely on the individual level. Other authors have argued that sport for development programmes would benefit from a more active policy advocacy role [56], and work elsewhere has documented various potential pathways for civil society organisations to engage in such policy advocacy [57]. Given the structural challenges around poverty, lack of employment opportunities, and immigration faced by the SAME participants, such advocacy could certainly be beneficial in the context of this sport-for-employability programme. Though beyond the scope of this paper, tactics such as public mobilisation, engaging with decision makers, or building coalitions with other organisations could be potentially valuable approaches [57].

## 7. Limitations and Conclusions

There are certainly limitations in the above analysis. As an external researcher engaging in an evaluation, there are potential limits to the depth of the above analysis. Longer-term locally led or supported research could help unearth more findings and nu-

ance. Likewise, employability programmes more broadly, and SAME in particular, would likely benefit from longer-term follow-up to check in on participants or analyse any changes made to programmes.

Recognising these limitations, this study nonetheless illustrates how programme participants face continued precarity, and this despite the fact that the SAME programme was widely appreciated and seen as supportive. Indeed, the level of personal support provided by SAME is unusual in the sport-for-employability space, and, in combination with its intensive, one-year programme, has the potential to form the basis of an improved model of sport-for-employability programming that works beyond the mere individual, or supply-side, factors. To reach this full potential, however, closer alignment with job market needs and engagement with policy issues are likely needed.

**Author Contributions:** Conceptualization, L.M., V.R., Y.M. and K.P.; methodology, L.M. and V.R.; formal analysis, L.M.; investigation, L.M.; resources, Y.M.; data curation, L.M.; writing—original draft preparation, L.M.; writing—review and editing, L.M., V.R., Y.M. and K.P.; visualization, L.M.; supervision, K.P.; project administration, L.M., V.R., Y.M. and K.P.; funding acquisition, K.P. All authors have read and agreed to the published version of the manuscript.

**Funding:** This research was funded as part of an evaluation on behalf of the German Development Corporation (GIZ) under project number 19.2005.7-006.00.

**Institutional Review Board Statement:** Ethical approval for this study was obtained by the German Sport University under application number 027/2022.

**Informed Consent Statement:** Informed consent was obtained by all participants in this study.

**Data Availability Statement:** Not applicable.

**Acknowledgments:** Thank you to the staff and participants of TIBU, who opened their doors and provided tremendous support for this research.

**Conflicts of Interest:** The authors declare no conflict of interest.

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
