# Peer review of "From Sport to Work? Exploring Potentials in a Moroccan Sport-for-Employability Programme"

_2673-995X, doi:10.3390/youth2040054_

Round 1
Reviewer 1 Report
In my opinion, the article submitted for evaluation meets the criteria of a reliable publication - a report on own research.
The theoretical context of the assumed and precisely formulated research problem was presented. The methodological concept based on qualitative research is clear and legitimate. In this sense, the research seems to be reliable and technically correct. The conclusions drawn (remembering the limitations of generalizations) are justified in relation to the presented results. The above opinion, as well as the assessments expressed in the form, are primarily of a formal nature. It should be remembered that any qualitative research in the field of social sciences also requires taking into account the social and cultural contexts of the territory. The cultural and social conditions of Morocco, and above all Casablanca (a distinction historically, culturally, economically and socially justified in the context of the diversity of Morocco known to me) are different from, among others, European. Hence, some nuances and threads may not be fully understood and obvious for the reviewer.
Reviewer 2 Report
This paper uses a qualitative approach to investigate the experiences and outcomes associated with an 45 employability-focused SFD programme based in Casablanca, Morocco
The research question is surely interesting, and I find the paper excellent.
The question is original, and results provide a certain advance in current knowledge. Results are commented in a proper way; hypotheses correctly identified; the Basic conceptual model presented in the paper is based on a proper literature.
The article is written in a perfect English and I think that conclusion perfectly fit the topic of the Journal.
Finally, I must say that this is one of the best papers I had the opportunity to revise in the last months, and I can recommend publication.
Reviewer 3 Report
Due to the course of action adopted, the work represents a very limited scientific slice for use in future research work. On the other hand, the presented results give an insight into the presented research in a clear and understandable way. The arguments cited constitute a kind of case study for further use by the author(s). The arguments adopted are scientifically correct and I have no objections to the methodological side. The bibliography used is adequate. It is worth noting that as many as 47% of the items are from the last 5 years (2018 and newer).
Reviewer 4 Report
Dear authors,
I have really enjoyed reading the paper and given its quality, I consider that I have little to contribute to it.
Congratulations for this excellent work.
Best regards,
The reviewer
